# Exfoliation of natural van der Waals heterostructures to a single unit cell thickness

Matěj Velický[1], Peter S. Toth[1], Alexander M. Rakowski[2], Aidan P. Rooney[2], Aleksey Kozikov[3], Colin R. Woods[3], Artem Mishchenko[3], Laura Fumagalli[3], Jun Yin[3], Viktor Zólyomi[3], Thanasis Georgiou[4], Sarah J. Haigh[2], Kostya S. Novoselov[3] & Robert A.W. Dryfe[1]

Weak interlayer interactions in van der Waals crystals facilitate their mechanical exfoliation to monolayer and few-layer two-dimensional materials, which often exhibit striking physical phenomena absent in their bulk form. Here we utilize mechanical exfoliation to produce a two-dimensional form of a mineral franckeite and show that the phase segregation of chemical species into discrete layers at the sub-nanometre scale facilitates franckeite's layered structure and basal cleavage down to a single unit cell thickness. This behaviour is likely to be common in a wider family of complex minerals and could be exploited for a single-step synthesis of van der Waals heterostructures, as an alternative to artificial stacking of individual two-dimensional crystals. We demonstrate p-type electrical conductivity and remarkable electrochemical properties of the exfoliated crystals, showing promise for a range of applications, and use the density functional theory calculations of franckeite's electronic band structure to rationalize the experimental results.

[1] School of Chemistry, University of Manchester, Oxford Road, Manchester M13 9PL, UK. [2] School of Materials, University of Manchester, Oxford Road, Manchester M13 9PL, UK. [3] School of Physics and Astronomy, University of Manchester, Oxford Road, Manchester M13 9PL, UK. [4] Manchester Nanomaterials Ltd, 83 Ducie Street, Manchester M1 2JQ, UK. Correspondence and requests for materials should be addressed to M.V. (email: matej.velicky@manchester.ac.uk) or to R.A.W.D. (email: robert.dryfe@manchester.ac.uk).

The research on two-dimensional (2D) materials has so far been mainly focused on unary and binary crystals such as graphene and MoS$_2$ (ref. 1) and little attention has been paid to more complex layered materials[2]. However, preferential phase segregation strongly dependent on chemical composition, which is a phenomenon that has previously been observed in ternary sulphides such as PbSnS$_2$ (ref. 3), could lead to the formation of layered structures and van der Waals heterostructures. Franckeite is a natural, thermodynamically stable, mixed-metal sulphide mineral, composed of lead, tin, antimony, iron and sulphur, first discovered in 1893 (ref. 4). It exhibits a distinctly layered structure, which is related to its complex chemical composition. The band gap of the individual metal sulphides, which combine to constitute franckeite, ranges from 0.37 eV in PbS (galenite) to 2.1 eV in SnS$_2$ (berndtite)[5], with the band gap of franckeite itself being previously determined by diffusive reflectance spectroscopy as 0.65 eV (ref. 6). Many other complex sulphides exist, with the band structure depending on their exact chemical composition and structure[6]. This offers great opportunities in band gap engineering[7], phase engineering[8], thermoelectric materials[3] and solar control coatings[9], which could all be realized through synthesis of complex metal-sulphides with on-demand properties.

Here we show that franckeite is a natural heterostructure exhibiting phase segregation into discrete layers held together by van der Waals forces, which facilitates its basal cleavage. We use scanning electron microscopy (SEM), transmission electron microscopy (TEM), energy-dispersive X-ray spectroscopy (EDXS) and X-ray photoelectron spectroscopy (XPS) to determine franckeite structure and chemical composition. Importantly, we show that franckeite can be exfoliated to a single unit cell thickness (1.85 nm), resulting in a high ratio between the lateral size and thickness of the exfoliated crystals, as confirmed using optical microscopy, atomic force microscopy (AFM), and Raman spectroscopy. The electronic transport measurements reveal that franckeite is a p-doped degenerate semiconductor and the electrochemical measurements show that it has a high intrinsic electric double-layer capacitance showing promise in energy storage applications. The density functional theory (DFT) calculations of franckeite's electronic band structure indicate only weak interactions between the individual van der Waals layers, also confirmed by the independence of franckeite's Raman spectrum of the number of layers and the incommensurate lattice matching observed by the high-angle annular dark-field (HAADF) scanning transmission electron microscopy (STEM).

## Results

### Morphology and chemical composition of bulk franckeite.
The SEM images in Fig. 1a–c reveal franckeite's layered nature, which facilitates its facile mechanical exfoliation. In the highest magnification image (Fig. 1c), terraces of micro-/nanoscopic width are clearly visible. The TEM-EDXS elemental maps in Fig. 1d–h show that the main elements, lead, tin, antimony, iron and sulphur, are homogeneously distributed when viewed perpendicular to the layers (along the [001] direction). The averaged EDXS spectrum in Fig. 1i and quantification in Table 1 were the basis for the compositional stoichiometry analysis, resulting in an approximate chemical formula of Pb$_{6.0}$Sn$_{3.1}$Sb$_{2.5}$Fe$_{1.1}$S$_{12.0}$O$_{1.1}$. This indeed best matches franckeite, a member of a complex group of metal sulphide minerals also including cylindrite, potosíite and incaite, which are found in the southwest of Bolivia, and have a generic chemical formula $(Pb, Sn)_{6+x}^{2+}Sb_2^{3+}Fe^{2+}Sn_2^{4+}S_{14+x}$, where $-1 \leq x \leq 0.25$ (ref. 10). The approximate chemical formula determined from the EDXS is not completely charge-balanced (with ca. 5–10% excess positive

charge, depending on the exact oxidation states of the metals) and it also does not fully fit the generic chemical formula of franckeite. This is mainly caused by the low sensitivity of the EDXS, the overlap between S K$\alpha$ and Pb M$\alpha$ peaks, and the presence of O, C and Ag impurities. Further SEM characterization and full TEM-EDXS quantification are found in Supplementary Note 1, Supplementary Fig. 1 and Supplementary Table 1.

### Crystal structure and phase segregation of franckeite.
Figure 2 summarizes the high-angle annular dark-field scanning transmission electron microscopy characterization of franckeite. The structure of this complex layered misfit compound mineral has troubled mineralogists, crystallographers and electron microscopists for several decades[11–13], until significant advances in understanding of its exact structure have eventually been achieved[14–16]. Our results confirm phase segregation into discrete Sn-rich, pseudo-hexagonal (H) and Pb-rich, pseudo-tetragonal (T) layers with a van der Waals gap between them. A cross-section through a thin exfoliated franckeite crystal was extracted to reveal a unit cell height of $\sim$1.85 nm (Fig. 2b,e). The two layers, H and T, are incommensurate with differing lattice dimensions in the [010] direction, which results in a mismatch periodicity of $\sim$4.3 nm visible as moiré fringes when viewed along the [001] direction in the TEM. This is also confirmed by the relative atomic displacements of the two layers in the cross-section (see Supplementary Note 2 and Supplementary Figs 2 and 3). Atomic-resolution HAADF-STEM (Fig. 2e) and STEM-EDXS elemental analyses of this structure (Fig. 2c,f) reveal significant compositional segregation at the atomic scale. Specifically, Pb in the inner two atomic planes of T layers is partially replaced by Sb, resulting in an alternating Pb-Sb-Pb arrangement (see the structure model in Fig. 2d). Small amount of Sn in the T layer ($\sim$8–10 at%) could indicate a partial replacement of Pb and/or Sb atoms by Sn. The average composition for the two layers individually is determined as Pb$_{6.3}$Sb$_2$S$_9$ (T) and Sn$_{2.3}$FeS$_{5.6}$ (H), resulting in an overall average chemical formula of Pb$_{6.3}$Sn$_{2.3}$Sb$_{2.0}$FeS$_{14.6}$.

Interestingly, the cross-sectional STEM images also show stacking faults, consisting of irregular lateral displacements of the SnS$_2$ layers along the [010] direction, which is likely to cause the forbidden lattice spots appearing in the electron diffraction data (see Supplementary Note 2 and Supplementary Figs 2 and 3). The thinnest region cross-sectioned for STEM analysis was $\sim$6 nm (or $\sim$3 unit cells) thick with the upper (freshly cleaved) surface terminating at the H layer and being covered with $\sim$0.5 nm thick layer of carbonaceous adsorbates (Fig. 2e). Surprisingly, the lower, substrate-bound surface terminates half-way through the Pb-rich layer. It is likely that such imperfect cleavage may be associated with a stacking fault, which locally weakens the structure. Such anomalies have also been confirmed by the observation of terraces with a sub-monolayer height. The structural analysis above suggests that the incommensurate stacking of the individual H and T layers (and therefore weak interaction between them) leads to a predominantly uniaxial basal cleavage of franckeite and a weak or no dependence of the electronic and structural properties of franckeite on the number of layers (see below).

Other related minerals such as SnS and SnS$_2$ have been recently shown to be readily exfoliated[17,18] and preferential elemental segregation to a single Pb-rich atomic plane within a PbSnS$_2$ lattice was also previously observed[3]. The phase segregation observed here, which gives rise to a van der Waals gap between the H and T layers, proves the concept of a naturally occurring heterostructure material, a phenomenon, which is likely to be

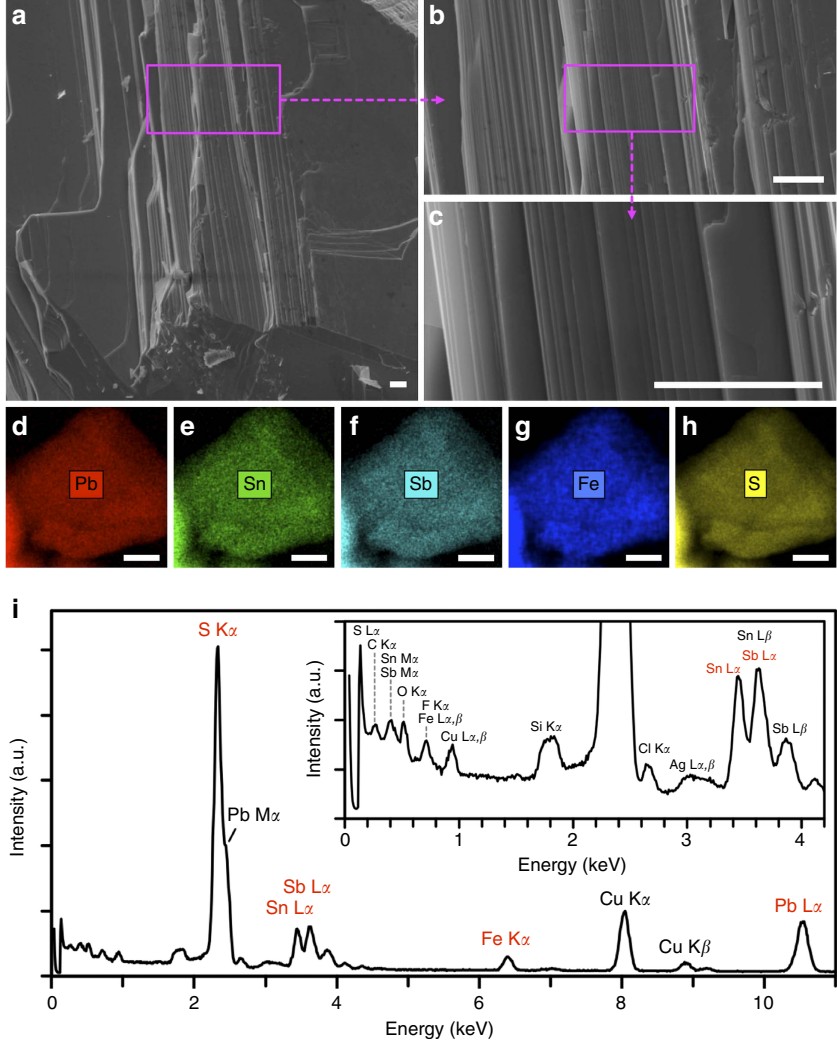

**Figure 1 | SEM and TEM-EDXS characterization of franckeite. (a–c)** SEM images of franckeite immobilized on a conductive carbon support. Zoom areas are highlighted by the magenta rectangles. The scale bars denote 5 μm. **(d–h)** TEM-EDXS maps of a franckeite crystal showing lead, tin, antimony, iron and sulphur, respectively. The scale bars denote 30 nm. **(i)** Averaged EDXS spectra with the inset showing the low-intensity peaks at low energies. The peaks of the five major elements used for the quantification shown in Table 1 are marked in red. The secondary peaks and the peaks originating from the substrate and impurities, which were de-convoluted and excluded from the quantification, are marked in black.

common among many complex minerals. This could be exploited simultaneously with the emerging advances in synthesis of metal chalcogenides and engineering of their properties[19], to create man-made van der Waals heterostructures with properties tailored to specific applications.

**Surface chemical composition of franckeite.** The surface sensitivity and a few-nanometre penetration depth of the XPS technique were utilized to determine the surface chemical composition of franckeite including the atomic oxidation states and the extent of surface impurities. Figure 3a shows an XPS spectrum of franckeite used for the quantification of the major elements shown in Table 2. The high-resolution spectra of Pb 4$f$, Sn 3$d$, Sb 3$d$ and S 2$p$ binding energy regions are shown in Fig. 3b–e. Compositional stoichiometry analysis yields a surface chemical formula of $Pb_{6.0}Sn_{2.0}Sb_{2.4}S_{13.8}O_{1.0}$. Peak fitting was used to determine the oxidation states of individual elements. Lead and tin (Fig. 3b,c, respectively) are present as $Pb^{2+}$ and $Sn^{2+}$ (both ~80%), and $Pb^{4+}$ and $Sn^{4+}$ (both ~20%). Antimony (Fig. 3d) is present as $Sb^{3+}$ (~60%) and $Sb^{5+}$ (~40%). Sulphur (Fig. 3e) is present exclusively as $S^{2-}$ and oxygen is bound both in metal

oxides ($O^{2-}$) and carbonaceous adsorbates. In comparison to the bulk-sensitive EDXS, only a negligible amount of Fe (<0.5 at%) and relatively lower concentration of Sn are detected from the surface-sensitive XPS. This suggests that the Sn- and Fe-rich H layers (Fig. 2c,f) are underrepresented on the surface of franckeite, possibly due to the surface being unstable upon exfoliation and degrading in air. Stoichiometric analysis, full quantification, and comparison of the freshly cleaved and air-aged crystals can be found in Supplementary Note 3, Supplementary Figs 4 and 5, and Supplementary Tables 2 and 3.

**Table 1 | EDXS quantification of bulk franckeite.**

| Element | Concentration (at%) |
|---|---|
| Pb | 24.1 ± 4.9 |
| Sn | 12.5 ± 2.5 |
| Sb | 10.2 ± 2.1 |
| Fe | 4.6 ± 0.4 |
| S | 48.6 ± 3.1 |

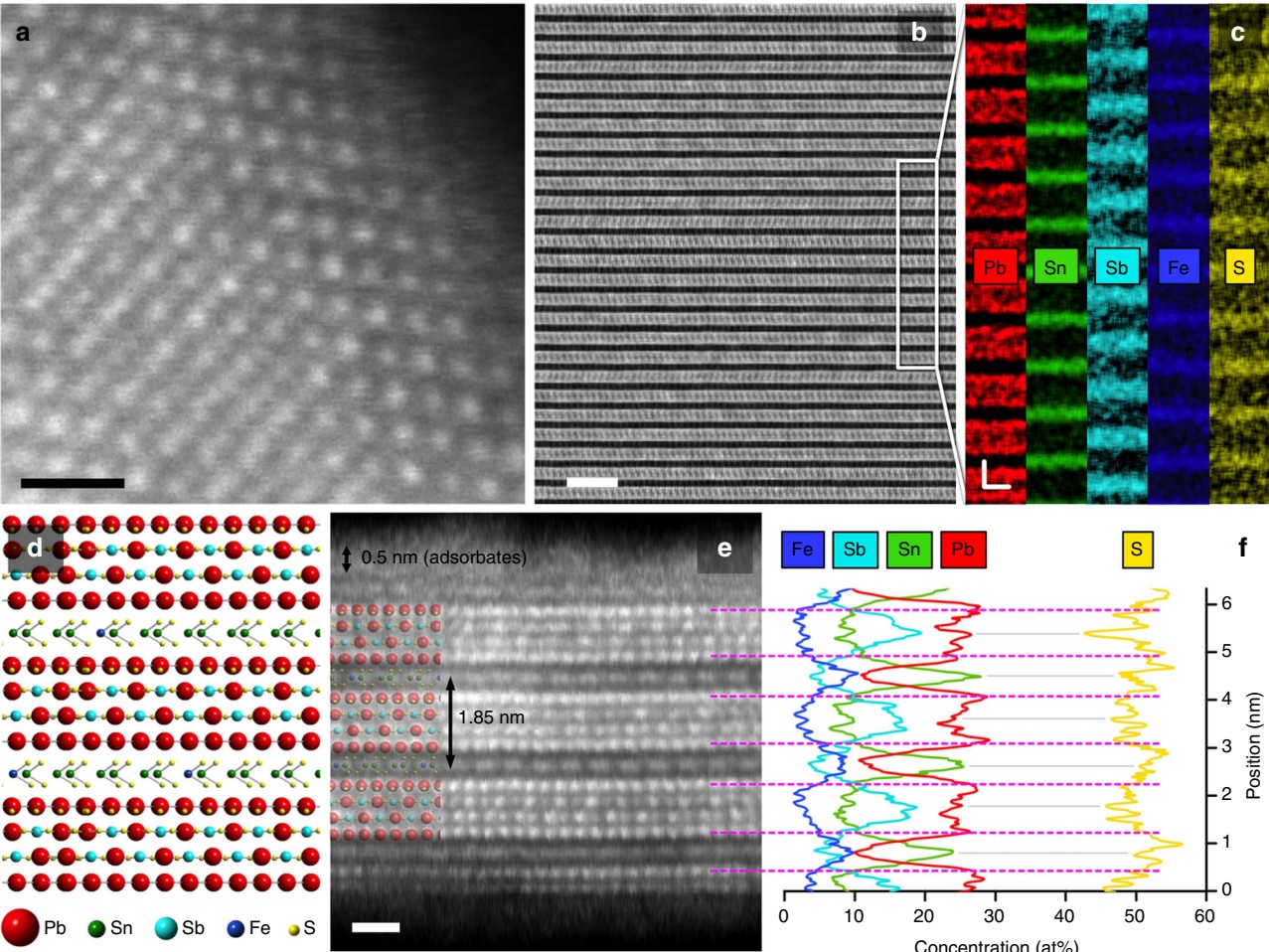

**Figure 2 | HAADF-STEM imaging of franckeite.** (**a**) High-resolution image of the crystal lattice viewed along the [001] direction (plan view of the basal plane). The scale bar corresponds to 1 nm. (**b**) Cross-sectional image of the layering viewed along the [100] direction. The scale bar corresponds to 5 nm. (**c**) Corresponding EDXS elemental intensity maps with the two-axis scale bar corresponding to 1 nm. (**d**) Proposed structure model of franckeite, with the relative size of atoms corresponding to their brightness in HAADF. (**e**) Atomic-resolution image of the cross-section for a few-layer crystal. The inset shows overlaid franckeite structure, the scale bar corresponds to 1 nm. (**f**) Corresponding EDXS concentration profiles showing compositional variation both between and within the individual layers. In contrast, no compositional variations are detected for the crystal viewed along the [001] direction. The pink dashed lines are a guide for the eye correlating the EDXS profile with the cross-sectional HAADF. The grey solid lines are a guide for the eye correlating the concentration peaks (Pb and Sn) and troughs (S).

**Preparation and characterization of thin franckeite.** Importantly, we succeeded in isolating monolayer (single unit cell thick) franckeite crystals. Thin franckeite layers were mechanically exfoliated onto an SiO2 substrate and characterized using optical microscopy, AFM and Raman spectroscopy. Bright-field and dark-field optical images of a monolayer franckeite crystal with adjacent few-layer and bulk crystals are shown in Fig. 4a,b. Corresponding AFM image and the height profile of the monolayer region are shown in Fig. 4c,d. The monolayers have typical thicknesses of 2.4–3.5 nm, which is in good agreement with the single unit cell repeat period determined from cross-sectional TEM (~1.85 nm), considering the additional increase in height originating from the AFM instrumental offset, layer of carbonaceous adsorbates, and impurities or moisture trapped between the crystal and substrate, which are known to beset characterization of 2D materials[2,20,21]. Furthermore, small terraces with less than a monolayer thickness were occasionally observed (Supplementary Fig. 6). Through mechanical exfoliation of a large number of crystals, we have discovered that the length of the few-layer crystals is

typically tens of μm, but their width rarely exceeds 0.5–1 μm, resulting in characteristic, needle-like, thin crystals with a large length-to-width aspect ratio (Fig. 4a-c and Supplementary Fig. 7). This preferential uniaxial basal cleavage is supported by the commensurate and incommensurate lattice stacking in the [100] and [010] directions, respectively, observed in the TEM/STEM.

Raman spectra of monolayer, few-layer and bulk franckeite obtained using a 532 nm laser excitation wavelength are shown in Fig. 4e (AFM was used to determine the thickness of individual franckeite flakes). Due to the small lateral size of thin franckeite crystals, only well isolated flakes were used for Raman spectroscopy measurements to avoid signal overlap with any adjacent thicker regions. Low laser power density was used to avoid franckeite degradation and a large number of individual measurements (14–28) at different locations on each flake were accumulated to build up each spectrum (substrate-uncorrected spectra are shown in Supplementary Fig. 8).

Despite the lack of Raman literature on franckeite, we make a tentative assignment of the main franckeite peaks based on the

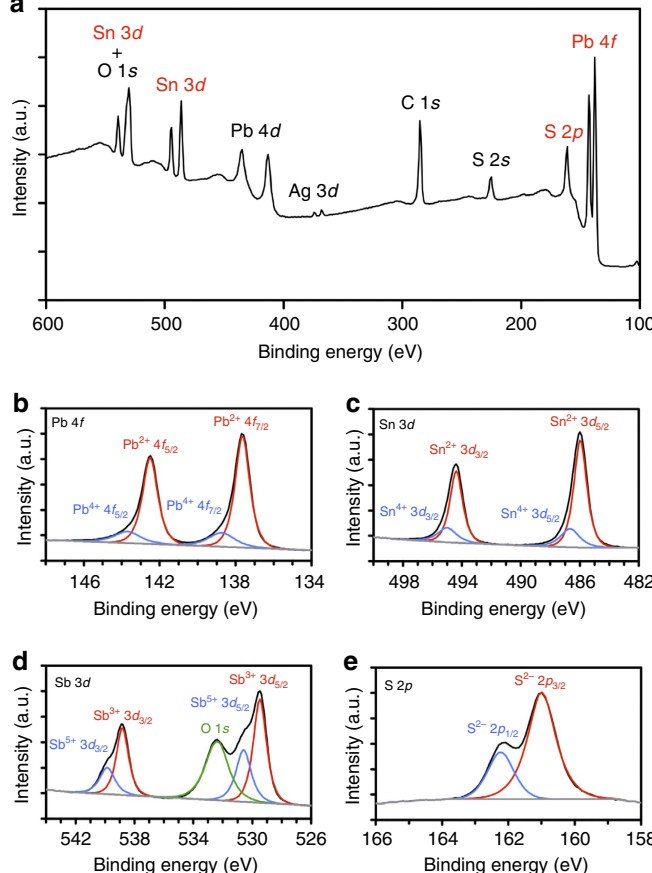

**Figure 3 | XPS characterization of franckeite surface.** (**a**) Average XPS spectrum of franckeite obtained from multiple individual measurements, referenced to the adventitious carbon C 1s peak at 284.7 eV (ref. 46). The peaks labelled in red were used for the quantification shown in Table 2. Other significant peaks, which were not used in quantification, are labelled in black. (**b**–**e**) High-resolution spectra of Pb 4f, Sn 3d, Sb 3d (+ O 1s), and S 2p binding energy regions. The intensities are normalized to the most intense peak within the respective spectral region.

**Table 2 | XPS quantification of franckeite surface.**

| Element | Concentration range (at%) | Average concentration (at%) |
|---|---|---|
| Pb | 23.5–26.8 | $24.7 \pm 0.3$ |
| Sn | 7.1–9.0 | $8.1 \pm 0.2$ |
| Sb | 9.0–13.1 | $10.1 \pm 0.2$ |
| S | 51.7–60.1 | $57.0 \pm 0.5$ |

available Raman data for the individual metal sulphides. We infer that the Raman spectrum of franckeite is dominated by $Sb_2S_3$ (stibnite) vibrations with a corresponding peak at $260\,cm^{-1}$ and a shoulder at $276\,cm^{-1}$ (ref. 22). $SnS_2$ (berndtite) vibrations are most likely to produce the peak at $320\,cm^{-1}$ (ref. 23). Both $Sb_2S_3$ and $SnS_2$ have large optical band gaps of 1.72 and 2.10 eV (ref. 5), respectively, and therefore exhibit comparatively weaker light absorption and hence stronger Raman scattering than PbS (0.37 eV)[5], which is a relatively weak Raman scatterer and does not contribute to the signal significantly[24]. A weak broad band, centred around $195\,cm^{-1}$ in bulk franckeite, which is hardly observable in monolayer could originate from SnS (1.01 eV) vibrations[5,25]. No significant changes in the

frequency of these Raman modes are observed for different franckeite thicknesses, which is to be expected due to the incommensurate stacking of the T and H layers. There is, however, a noticeable increase in the Raman intensity for bilayer and trilayer in contrast to monolayer and bulk (Fig. 4d). Such enhancement has previously been observed for other 2D materials including graphene[26], $MoS_2$ (ref. 27) and $MoSe_2$ (ref. 28), and it can be explained by an optical interference in the 2D crystal/$SiO_2$/Si system and corresponding optical field enhancement for certain 2D crystal thicknesses[29]. Furthermore, no photoluminescence was detected within a 532–900 nm range for monolayer or bulk (Supplementary Fig. 9), which agrees with the indirect mid-/far-infrared band gap of franckeite determined by the transport measurements and predicted by the DFT.

Raman spectroscopy data suggest that the electronic and structural properties of franckeite do not depend on the number of layers, which is to be expected given the incommensurate stacking and hence weak interaction between the T and H layers. Importantly, the phase segregation into discrete van der Waals layers, which is likely to be a common denominator for a wider family of naturally occurring complex minerals, could be utilized for preparation of man-made van der Waals heterostructures as a formidable alternative to their painstaking construction by stacking of the individual 2D materials on top of each other[30]. Furthermore, thin franckeite shows remarkable thermodynamic stability after exfoliation in air up to more than 6 months, as confirmed by optical microscopy, AFM, Raman spectroscopy. Transport and electrostatic force microscopy (EFM) measurements, which are more sensitive to crystal degradation, show that all but monolayer franckeite flakes are conductive. Such stability is exceptional in comparison to some other novel 2D materials, such as phosphorene or $NbSe_2$, which often degrade in air when isolated in their mono- or few-layer form[31]. Further characterization using optical microscopy, AFM, and Raman spectroscopy, including laser-induced degradation and surface ablation is detailed in Supplementary Note 4 and Supplementary Figs 10 and 11.

**Electrochemistry and liquid-phase exfoliation of franckeite.** To explore the potential use of franckeite in energy storage and conversion applications, we have determined its electrochemical properties using a micro-droplet cell measurement, representative results of which are summarized in Fig. 5. This approach was previously applied to topography-dependent electrochemical measurements on graphene and $MoS_2$ (refs 20,32,33). Since the cleavage of franckeite is mostly limited to a single plane, we use the term 'basal' here to describe its low-defect (001) surface, and term 'edge' to describe other surfaces with an increased density of edges, terraces, and defects. We have found that the average electric double-layer capacitance measured by cyclic voltammetry on the basal surface is $27.4 \pm 2.2\,\mu F\,cm^{-2}$ and further increases to $153 \pm 62.8\,\mu F\,cm^{-2}$ on the edge surface (Fig. 5a). We argue that this is directly related to franckeite's electronic structure as it has been previously proposed for $MoS_2$ that steric accessibility of metallic orbitals at the crystal edges increases its electrochemical performance[34]. In comparison, typical capacitance values measured using the same method are ca. $15$–$30\,\mu F\,cm^{-2}$ for unpolished platinum, $2$–$4\,\mu F\,cm^{-2}$ for the basal plane of $MoS_2$, and $1$–$2\,\mu F\,cm^{-2}$ for the basal plane of graphene. It is important to note here that the basal plane surface of mechanically exfoliated 2D materials is very flat, unlike the disordered and porous surface of multi-flake materials, typically prepared by liquid-phase exfoliation. The real surface area of liquid-phase exfoliated 2D materials is often orders of magnitude

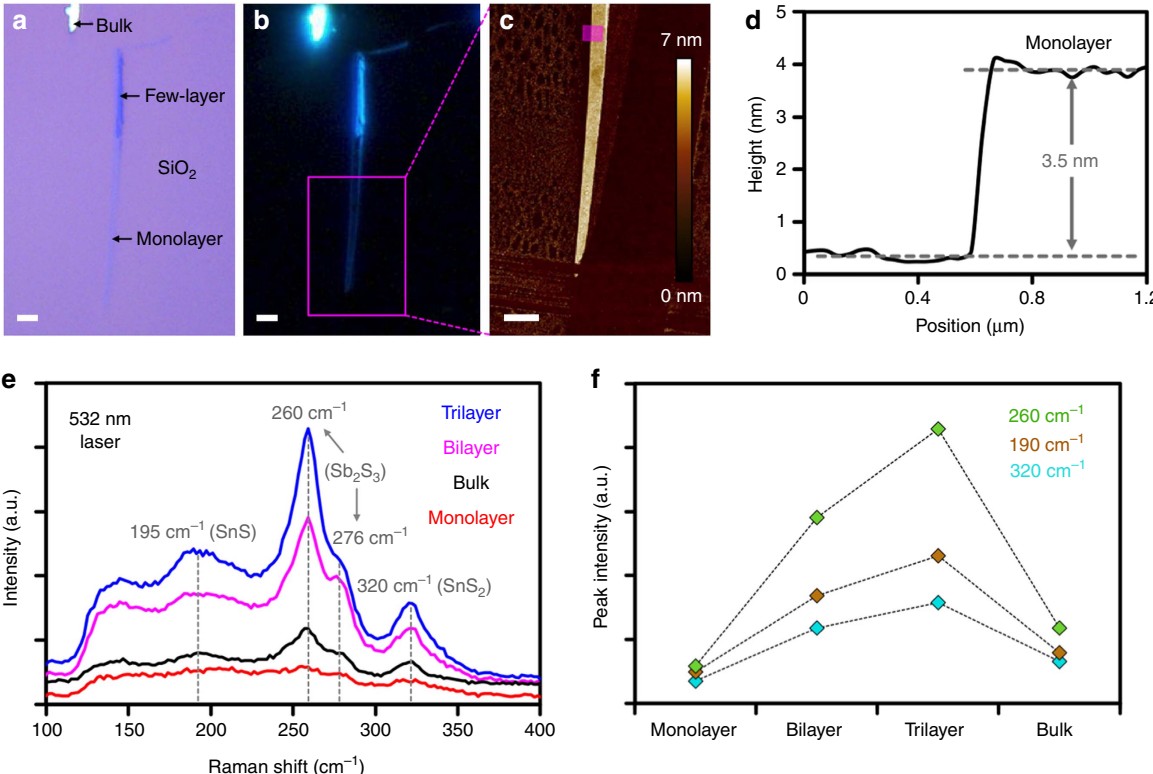

**Figure 4 | Microscopic and spectroscopic identification of thin franckeite.** (**a,b**) Bright-field and dark-field optical images of a franckeite crystal exfoliated on an $SiO_2$/Si substrate. (**c**) AFM image of a selected monolayer area indicated by the magenta rectangle in **b**. All scale bars denote 2 μm. (**d**) Step-height profile of a monolayer franckeite flake taken from an area indicated by the transparent magenta rectangle in **c**. (**e**) Raman spectra of monolayer, bilayer, trilayer and bulk franckeite using a 532 nm laser excitation wavelength at 19 kW cm$^{-2}$ power density. Spectrum of the underlying Si substrate is subtracted from the thin layer spectra ($\leq$3). (**f**) Intensity of the three major peaks as a function of thickness.

larger than the apparent surface area, resulting in artificially inflated capacitance values[35–37]. The high capacitance in our case therefore indicates a degenerate semiconducting nature of franckeite, also confirmed by independence of the voltammetric response of illumination intensity, and an additional contribution from a redox activity, which can be observed when the potential window is extended (see Supplementary Note 5 and Supplementary Figs 12 and 13). We cannot completely rule out that edges/defects on the surface, or even solution-induced delamination of the crystals, may contribute to the capacitance increase. However, this would in fact aid franckeite liquid-phase exfoliation and ion intercalation.

To demonstrate the feasibility of this approach, we perform ultrasonication-assisted liquid-phase exfoliation of franckeite in five different solvents, results of which are summarized in Fig. 5d–f. Similarly to other 2D materials, N-methyl-2-pyrrolidone (NMP) offers the highest concentration of the exfoliated material[38,39], followed by N,N-dimethylformamide (DMF) and acetone (Fig. 5d). No visible material remains in suspension after exfoliation in isopropyl alcohol (IPA) or water. A representative AFM image in Fig. 5e shows that the exfoliated flakes tend to re-stack into large agglomerates, although thin individual crystals with thickness less than five layers are still found (Fig. 5f). These results demonstrate that liquid-phase exfoliation could be used as an alternative, scalable method of thin franckeite production.

Heterogeneous electron transfer rate measurement using two common redox mediators, $Ru[(NH_3)_6]^{3+/2+}$ and $[Fe(CN)_6]^{3-/4-}$, is shown in Fig. 5b, yielding average values of the standard rate constant of $(0.62 \pm 0.34) \times 10^{-3}$ cm s$^{-1}$ for $Ru[(NH_3)_6]^{3+/2+}$ and $(0.90 \pm 0.17) \times 10^{-3}$ cm s$^{-1}$ for $[Fe(CN)_6]^{3-/4-}$. These

values exceed those determined on graphite and $MoS_2$ using the same experimental method[20]. Hydrogen evolution, an important technological reaction, was difficult to quantify in a diffusion-limited regime within the micro-droplet electrochemical cell. Nevertheless, some hydrogen evolution activity is observed on basal surface in 1 M HCl solution as compared with the blank electrolyte solution (Fig. 5c).

**Transport measurement of thin franckeite.** We also measured the electronic transport properties of devices with various franckeite thicknesses (selected device images are shown in Supplementary Fig. 14). Despite numerous attempts, we failed to observe conductivity in monolayer devices (six in total), most likely due to franckeite degradation caused by the loss of certain atomic species or surface oxidation as suggested by the XPS results. Bilayer devices, on the other hand, do reproducibly conduct. Figure 6 demonstrates the typical transport characteristics of bilayer and five-layer franckeite. Devices of both thicknesses display electric field-effect behaviour (Fig. 6a,b), with the bilayer device (Fig. 6c) showing stronger conductance modulation than the five-layer device (Fig. 6d). From the gate voltage dependence, we conclude that the material is a p-doped semiconductor. Current-bias voltage characteristics are non-linear, suggesting the formation of a Schottky barrier at the interface between the metal and franckeite, also supported by the strong temperature dependence of current–voltage characteristics (Fig. 6c). Additional data for a device of four-layer thickness are presented in Supplementary Note 6 and Supplementary Fig. 15. The Arrhenius plots of the zero-bias

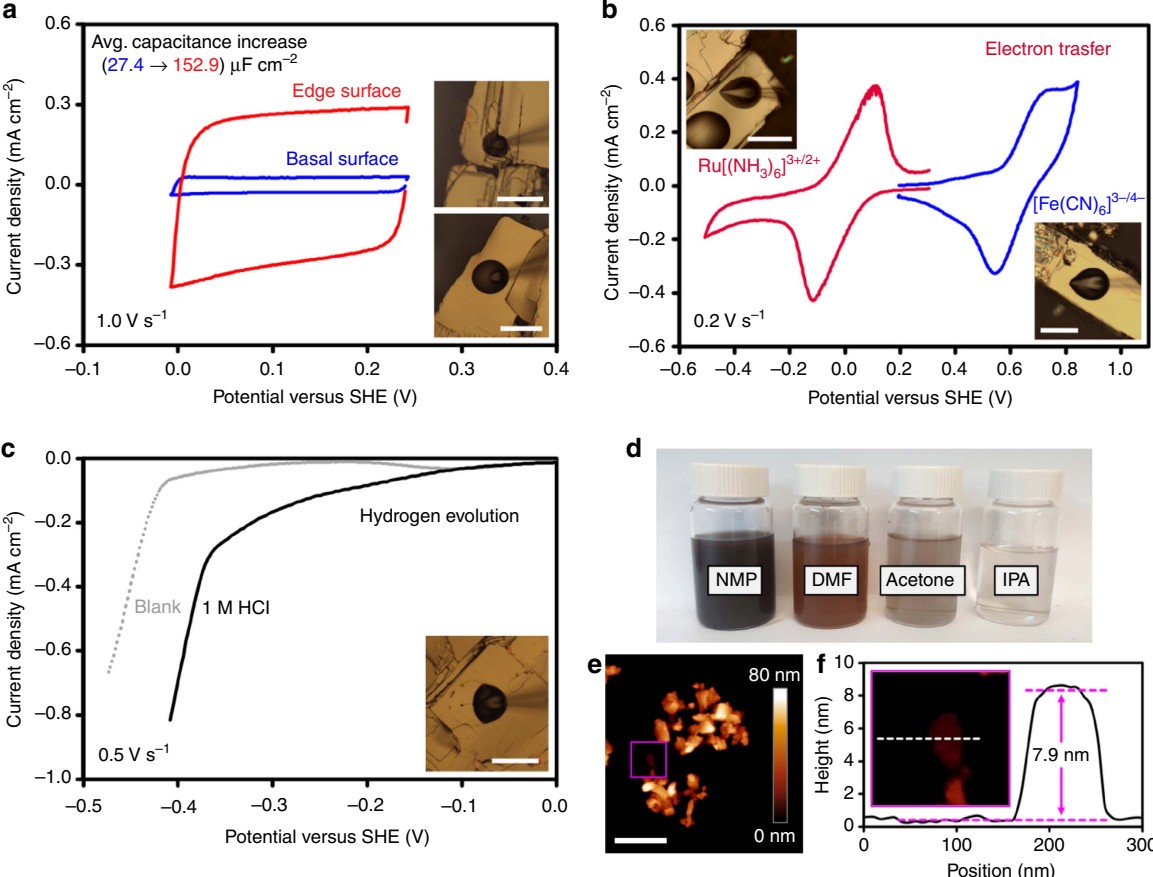

**Figure 5 | Electrochemical characterization and liquid-phase exfoliation of franckeite.** (**a**) Capacitance measurement on the basal and edge surface using a 6 M LiCl aqueous supporting electrolyte. (**b**) Electron transfer measurement on basal surface using $Ru[(NH_3)_6]^{3+/2+}$ and $[Fe(CN)_6]^{3-/4-}$ redox mediators in 6 M LiCl. (**c**) Hydrogen evolution measurement using 1 M HCl in 6 M LiCl. The optical images in the inset show the micro-droplet electrochemical cells utilized for the measurement, scale bars are 50 μm. (**d**) Photograph of franckeite suspensions in NMP, DMP, acetone and IPA, after an ultrasonication-assisted liquid-phase exfoliation of 5 mg ml$^{-1}$ franckeite solution and subsequent centrifugation. (**e**) AFM image of an agglomerate of thin franckeite flakes following a drop-cast transfer of the NMP suspension onto an SiO$_2$/Si substrate and subsequent solvent evaporation. The scale bar corresponds to 500 nm. (**f**) Height profile of a four-layer franckeite flake corresponding to the white dashed line in the inset AFM image.

conductance dependence on temperature (Fig. 6e) allowed us to extract the activation energies of 220 meV for bilayer, 170 meV for four-layer and 80 meV for five-layer devices. Such activation energies are comparable with the size of the band gap predicted by our DFT calculations (0.25–0.35 eV) but it is lower than the band gap determined previously by the diffuse reflectance spectroscopy (0.65 eV)[6]. We also verified the transport measurements results by performing EFM measurements on thin franckeite, and confirmed, that while the bilayer and thicker franckeite layers are conductive, the monolayers are insulating (see Supplementary Note 6 and Supplementary Fig. 16).

**Electronic band structure calculations**. To understand the transport, electrochemistry and Raman spectroscopy results, we performed DFT calculations of the electronic band structure of franckeite. First, we calculated the electronic structure of T and H layers using a simplified stoichiometry: Pb$_3$SbS$_4$ for the T layer and SnS$_2$ for the H layer. The resulting band structures in Fig. 7a (T layer) and Fig. 7b (H layer) reveal a very large difference in the work functions for the two layers (3.76 eV).

Furthermore, the T layer exhibits a small direct band gap of 0.36 eV at the C point, whereas the H layer has a large indirect band gap of 1.48 eV, with the conduction band minimum at the C point and the valence band maximum half way between the

Y and Γ points. The T–H heterostructure (franckeite monolayer) is modelled in a commensurate approximation, for which the band structure displays a prominent energy gap just below the Fermi level (Fig. 7c). This gap is indirect with the conduction band minimum at the Γ point and the valence band maximum at the C point, and its magnitude is 0.35 eV, which is almost identical with that of the T layer (0.36 eV). The separation between the mean planes of the T and H layers was found to be 0.9054 nm, which compares reasonably well with the value obtained experimentally from the HAADF-STEM (0.925 nm, Fig. 2). In practice, we expect that the crystal reconstruction and the moiré pattern formation should occur locally to accommodate a local van der Waals interaction, which is indeed observed in the TEM (Supplementary Fig. 2). An important consequence of the strongly incommensurate lattice parameters of the T and H layers is that the interaction between the layers should be weak and that the physical properties of franckeite should not depend strongly on the number of layers, which is indeed observed by the Raman spectroscopy.

The density of states for the monolayer (T–H), one and a half monolayer (T–H–T), bilayer (T–H–T–H) and bulk franckeite, are presented in Fig. 7d. The band gap is predicted to be around 250–350 meV and it decreases with increasing number of layers, which corresponds reasonably well with the observed gaps in our transport experiments, and also with the absence of

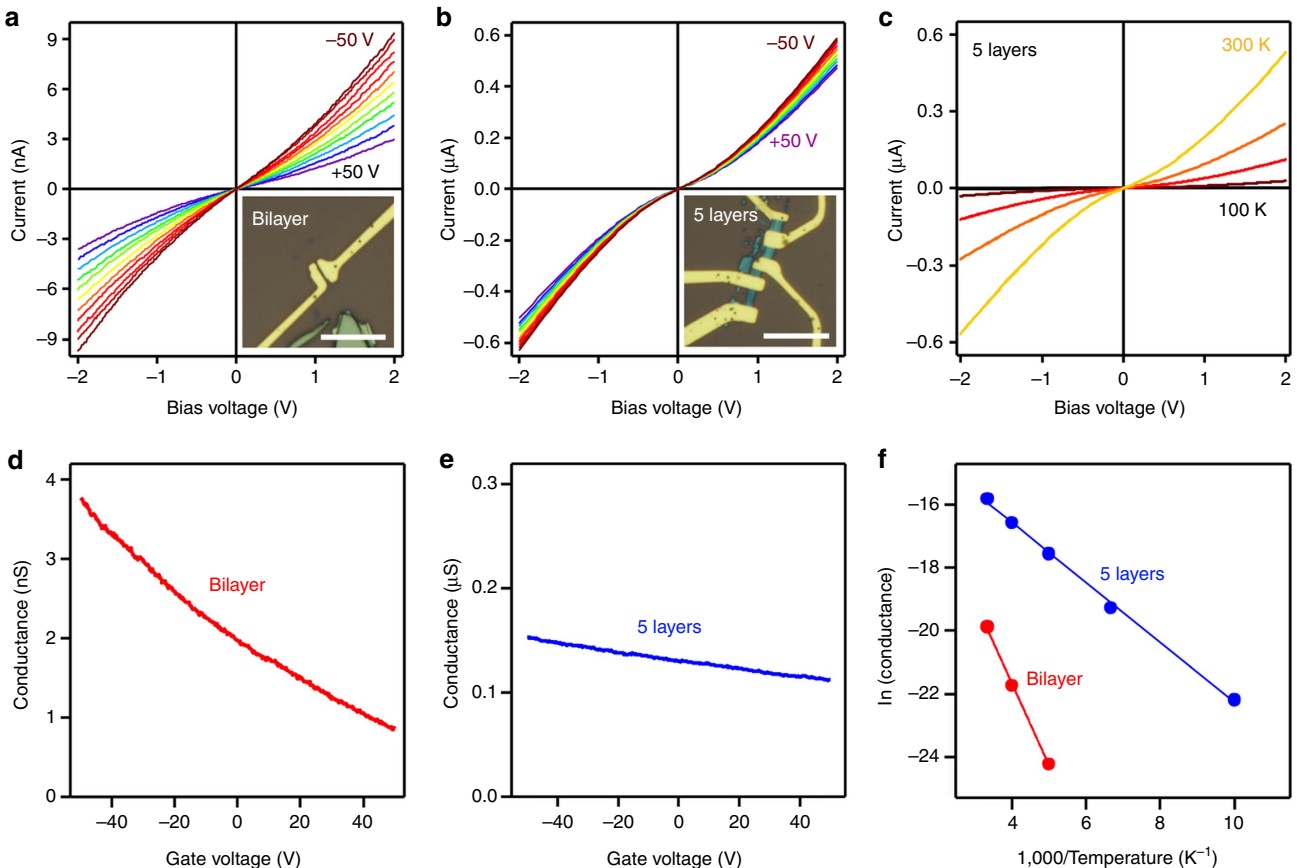

**Figure 6 | Transport characterization of franckeite.** (**a**) Gate voltage dependence of the current–voltage for bilayer franckeite device between − 50 and 50 V (brown to violet), in 10 V increments at 300 K. Bottom-right inset: an optical image of the device, the scale bar corresponds to 10 μm. (**b**) Same as **a** for a five-layer franckeite device. (**c**) Current–voltage curves for the five-layer device between 300 and 100 K (yellow to black), in 50 K increments. (**d**) Conductance dependence on gate voltage (300 K) for the bilayer device. (**e**) Same as **d** for the five-layer device. (**f**) Temperature dependence of the zero-bias conductance for the bilayer (red) and five-layer (blue) device, solid lines are Arrhenius fits.

photoluminescence in the visible spectrum. At the same time, our calculations predict the system to be n-doped, which is caused by the use of simplified stoichiometry. In reality, the presence of a small amount of Sn in the T layer and Fe in the H layer would have to be taken into account. These atoms will create donor and acceptor states in T and H layers of franckeite, respectively, strongly reducing the expected charge transfer between the layers once they are brought together. This drives the system towards a gapped semiconductor state due to the acceptor states countering the rise of the Fermi level in the H layer and the donor states countering the fall of the Fermi level in the T layer, which is energetically favourable. Further discussion of this complex behaviour is detailed in Supplementary Note 7.

## Discussion

In summary, we have succeeded in mechanical exfoliation of a mineral franckeite to a single unit cell thickness, which is facilitated through phase segregation into discrete layers at the nanometre scale. We show that this material exfoliates to thin, needle-like monolayer and few-layer crystals and we also demonstrate feasibility of its liquid-phase exfoliation in several solvents as a scalable production alternative. The key electrochemical properties of franckeite, capacitance and electron transfer rate, are considerably higher than those of more common 2D materials. In particular, the capacitance of

franckeite exceeds that of graphene and MoS₂ by one order of magnitude, which is promising for emerging energy storage technologies based on supercapacitors. The electronic transport measurements show that franckeite is a p-doped semiconductor with a mid-/far-infrared band gap, and the EFM measurements confirm that crystals of all thicknesses, except for the monolayer, are conductive. On the basis of the Raman spectroscopy, electron microscopy and DFT results we suggest that the electronic and structural properties of franckeite do not depend on the number of layers. The important findings of the current work are the role of the polar nature of the double-layer structure, charge transfer between the individual layers, and role of the impurities, which balance the large charge transfer. Even though it is yet unclear whether this particular material will be used in applications, these findings are an important milestone in studying the principles of formation and stability of a new class of complex van der Waals crystals, which can be utilized for preparation of man-made heterostructures.

## Methods

**Sample preparation.** Samples were prepared by the 'scotch-tape' mechanical exfoliation of natural franckeite crystals originating from Poopo, Oruro Department, Bolivia (Manchester Nanomaterials Ltd, UK) onto three different types of substrate: carbon adhesive discs for the SEM and XPS analyses, lacey carbon-coated copper grids for the STEM, TEM and EDXS analyses (both Agar Scientific, UK), and oxidized silicon wafers (IDB Technologies, UK) for the electrochemical measurements, Raman spectroscopy, transport measurements, EFM and focused ion beam (FIB) milling. Exfoliated crystals were stable in air for

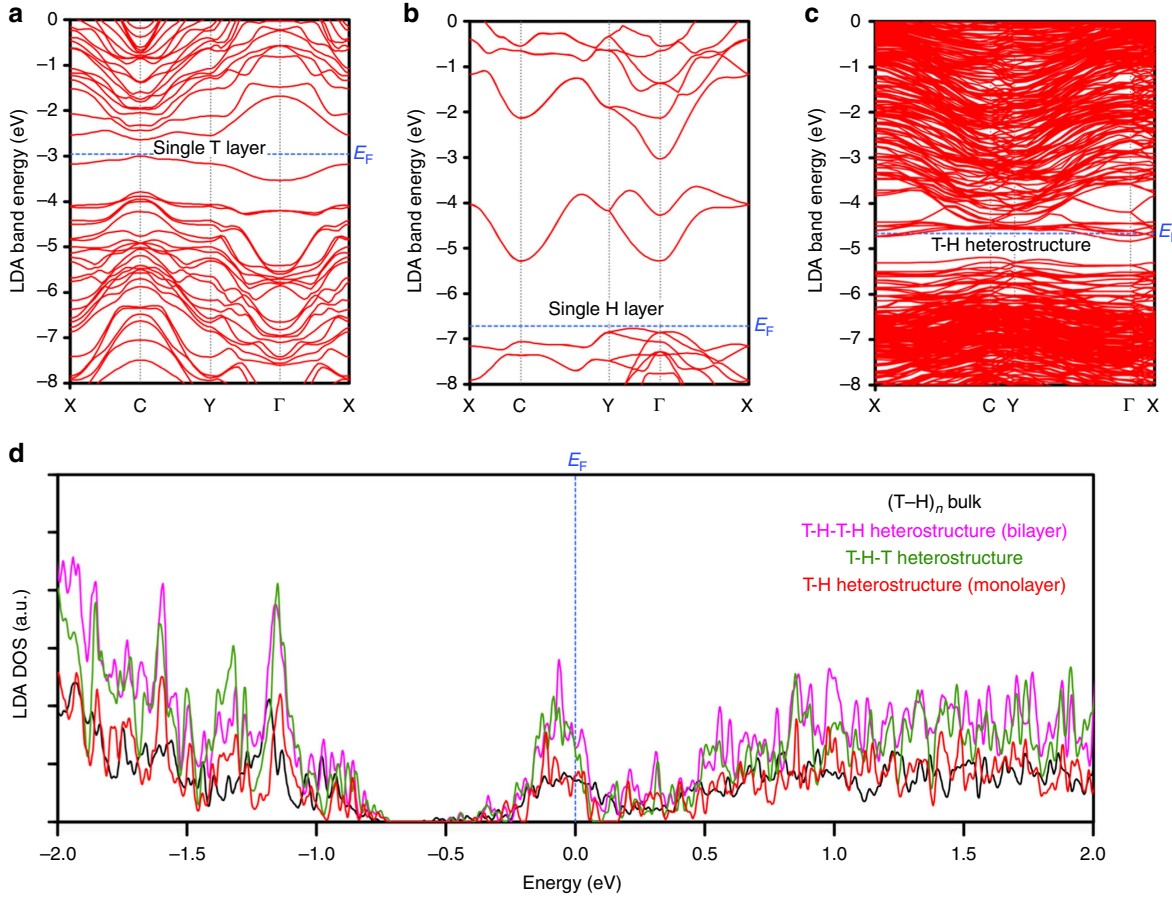

**Figure 7 | DFT calculations of franckeite's electronic band structure. (a–c)** Electronic band structure of the T layer, H layer and T–H heterostructure (franckeite monolayer), respectively, determined using the local density approximation (LDA) of DFT. The vertical grey dotted lines follow the high symmetry points in the Brillouin zone. **(d)** LDA density of states (DOS) for the T–H heterostructure (monolayer), T–H–T heterostructure, T–H–T–H heterostructure (bilayer) and bulk franckeite. The blue dashed lines in all panels correspond to the Fermi level.

at least 6 months, based on no observable changes in optical microscopy, Raman spectroscopy, and AFM, except for monolayer franckeite, which was not conductive in the transport measurements and EFM, indicating possible degradation. Two-terminal devices for the transport measurements were prepared using electron-beam lithography followed by evaporation of Cr/Au (4/50 nm) contacts (insets of Fig. 6a,b). Some of the devices were encapsulated in thin layers of hexagonal boron nitride (hBN) to protect franckeite from oxidation and contamination (Supplementary Figs 6 and 14). To achieve this, hBN was mechanically exfoliated onto an Si-supported polymethylglutarimide (PMGI)/poly(methyl methacrylate) (PMMA) substrate and then transferred on top of the selected franckeite flakes. The PMGI/PMMA stack was later dissolved using acetone. One-dimensional contacts to the hBN-encapsulated franckeite flake were made following previously described procedure[40], in which the first electron-beam lithography was used to define a mask on top of the hBN/franckeite heterostructure. The regions inside the mask were etched using an $O_2$/Ar plasma and the franckeite flake was connected using Cr/Au contact evaporation.

**SEM and XPS characterization.** SEM images were collected using Philips XL30 ESEM-FEG scanning electron microscope (FEI Company, USA) operating at 15 kV accelerating voltage. XPS spectra were obtained using a K-Alpha monochromated XPS spectrometer (Thermo Fisher Scientific Inc) and analysed using CasaXPS software v.2.3 (Casa Software Ltd).

**TEM/STEM/EDXS characterization.** Electron diffraction patterns were collected using a Philips CM20 TEM operated at 200 kV accelerating voltage. STEM images and EDXS elemental maps were obtained using a Titan G2 STEM (FEI Company, USA) operated at 200 kV, equipped with a Super-X EDX detector and GIF quantum energy filter. HAADF-STEM images were acquired with a convergence angle of 21 mrad, an inner angle of 54 mrad and a probe current of ∼75 pA, EDXS data were quantified using Esprit software version 1.9 (Bruker, USA). The crystals were aligned using Kikuchi bands in the $SiO_2$/Si substrate. Cross-sectional sample preparation was performed using

a dual FIB Nova NanoLab instrument (FEI Company, USA) fitted with an Omniprobe nano-manipulator (Oxford Instruments, UK). More details of the milling procedures can be found in Supplementary Note 2.

**Optical microscopy and Raman spectroscopy.** A Nikon Eclipse LV100ND optical microscope and a DS-Fi2 U3 CCD camera (Nikon Metrology, UK Ltd) were used to image franckeite flakes in both bright-field and dark-field illumination modes. Raman spectroscopy was measured with an inVia spectrometer, using either 532 or 633 nm laser excitation wavelength, and a ×100 objective (Renishaw plc, UK), resulting in a laser spot size of ca. 0.8 μm².

**AFM and EFM measurements.** AFM measurements were performed with a Bruker Dimension 3,100 V instrument in a tapping mode with a tip resonance frequency of ∼350 kHz. EFM measurements were carried out using a direct current (DC) bias voltage applied between doped silicon tip (Nanosensors PPP-FMR, 0.5–9.5 N m⁻¹) and the underlying doped silicon substrate.

**Electrochemical measurements.** Franckeite crystals were electrically contacted to a copper wire using a silver conductive paint (RS Components Ltd, UK). Aqueous micro-droplets of either pure 6 M LiCl electrolyte, 3 mM redox mediator ($Ru(NH_3)_6Cl_3$ or $K_3Fe(CN)_6$) in 6 M LiCl or 1 M hydrochloric acid in 6 M LiCl, deposited onto the flake surface using a pressure-controlled glass micropipette, were used as microscopic electrochemical cells. Ultra-pure deionized water (18.2 MΩ cm, Milli-Q Direct 8, Merck Millipore, USA) was used for preparation of these solutions. Electrochemical measurements were controlled by a PGSTAT302N potentiostat (Metrohm Autolab B.V., The Netherlands) and were carried out in a three-electrode configuration, employing the crystal surface as a working electrode, and platinum wire and silver chloride wire as a counter and a reference electrode, respectively. All measurements were performed at room temperature (25–28 °C) and the potential is referenced to the standard hydrogen electrode.

All chemicals were of 98% or higher purity and were purchased from Sigma-Aldrich, UK. Copper, silver and platinum wires (>99.9%) were purchased from Advent Research Materials, UK. Further details can be found in Supplementary Note 5, and in our previous publications[20,32,33].

**Electrochemical analysis.** The standard heterogeneous electron electron transfer rate constant, $k^0$, of the redox mediator oxidation/reduction was calculated from the following equation[41].

$$k^0 = 2.18 \left( \frac{\alpha n F D v}{RT} \right)^{0.5} e^{-\frac{\alpha^2 n F}{RT} \Delta E_P} \quad (1)$$

where $\alpha$ (assumed to equal 0.5 due to the reaction symmetry) is the transfer coefficient, $n$ is the number of electrons exchanged in the reaction ($n = 1$ for both redox mediators), $F$ is the Faraday constant, $D$ is the diffusion coefficient of the redox mediator, $v$ is the scan rate, $R$ is the universal gas constant, $T$ is the thermodynamic temperature and $\Delta E_P$ (larger than 220 mV) is the peak-to-peak separation of the redox mediator reduction/oxidation reaction. The Nicholson method, based on the following equation, was used for $\Delta E_P$ smaller than 220 mV (refs 42,43).

$$\psi = \frac{(-0.629 + 0.002 n \Delta E_P)}{(1 - 0.017 n \Delta E_P)} = k^0 \sqrt{\frac{RT}{\pi n F D}} v^{-0.5} \quad (2)$$

$\Delta E_P$ was measured for each micro-droplet for a range of scan rates (0.1–1.0 V s$^{-1}$) and the mean $k^0$ value determined using the equations (1) and (2).

The electric double-layer capacitance, $C_{EDL}$, in a 6 M LiCl aqueous supporting electrolyte was determined from cyclic voltammetry using the following equation[44].

$$C_{EDL} = \frac{1}{2v(E_{max} - E_{min})} \oint_E I(E) dE \quad (3)$$

where $E$ is the applied potential and $E_{max}(E_{min})$ are the maximum (minimum) potentials, which limit the voltammetric scan. The mean $C_{EDL}$ was determined from several measurements at different scan rates between 0.3–3.0 V s$^{-1}$.

**Liquid-phase exfoliation.** Millimetre-size, bulk crystals of franckeite were ground in a mortar and pestle. The resulting fine powder was loaded in 20 ml of NMP, DMF, acetone, IPA and water, respectively, at a fixed concentration of 5 mg ml$^{-1}$. The powder–solvent mixtures were sonicated in a PC620R-1 Bransonic ultrasonic bath for 1 h at constant temperature (15 °C). The resulting suspensions were then centrifuged twice at 3,000 r.p.m. for 20 min using a Corning LSE Compact centrifuge to remove thicker non-exfoliated material. The recovered supernatant was then transferred on an SiO$_2$/Si substrate by drop-casting and subsequent evaporation of the solvent for characterization by AFM.

**Transport measurements.** Standard two-terminal DC transport measurements were performed in a helium atmosphere using a variable temperature insert fitted into a He-4 cryostat. Sample temperature was controlled using an ITC503S temperature controller (Oxford Instruments). Current-bias voltage characteristics were measured at different gate voltages using a 2614b Keithley dual-channel source metre.

**Density functional theory calculations.** The optimal crystal structures of T and H layers have been calculated within the LDA of DFT using the VASP code[45]. The structure was approximated so that no Sn and Fe atoms were assumed to be present in the T and H layer, respectively. To calculate the electronic band structure of the T–H heterostructure (franckeite monolayer), a super-cell containing 128 atoms with a tolerance set to 0.05 nm was constructed. More details on the DFT calculations are found in Supplementary Note 7.

**Data availability.** The data that support the findings of this study are available from the University of Manchester at doi: 10.15127/1.306767.

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

## Acknowledgements

This work was supported by the EPSRC (grants EP/I005145/1, EP/K016954/1, EP/K016946/1, EP/M010619/1, EP/L01548X/1, EP/G03737X/1, EP/K000225/1 and EP/N007131/1), ERC, Graphene Flagship (contract no. NECTICT-604391), Royal Society, Defense Threat Reduction Agency (grant HDTRA1-12-1-0013), N8 HPC facilities, CSF cluster at the University of Manchester, US Army Research Office, US Navy Research Office, and US Airforce Research Office. The Titan was funded by H.M government UK. We also thank NEXUS at nanoLAB (Newcastle University) for the XPS measurement.

## Author contributions

M.V. conceived and designed the project, prepared and characterized the samples using the optical microscopy, Raman spectroscopy and electrochemistry, analysed the XPS data and wrote the manuscript, P.S.T. carried out the SEM and contributed to the Raman spectroscopy, S.J.H., A.P.R., and A.M.R. carried out the FIB, TEM, STEM and EDXS measurements and analysis, A.K. carried out the device fabrication and characterization using optical microscopy and AFM, C.R.W. contributed to the AFM measurements, A.M. and J.Y. performed and analysed the transport measurements, L.F. performed the EFM measurements, T.G. carried out the liquid-phase exfoliation, and V.Z. performed the DFT calculations. K.S.N., R.A.W.D. and all other authors contributed to the discussion and interpretation of the results.

## Additional information

**Competing financial interests:** The authors declare no competing financial interests.

