## [Peer Review File · Nature Communications]

Reviewers' Comments:

Reviewer #1 (Remarks to the Author)

In the manuscript titled: "Phase segregation facilitates exfoliation of franckeite crystals to a single unit cell thickness" by Matěj Velický and collaborators, a new atomically thin material is presented. The authors managed to isolate, by mechanical exfoliation, a single unit cell of a complex sulfide containing Lead (Pb), Tin (Sn), Antimony (Sb) and Iron (Fe), being Pb the one with highest atomic concentration (20 at %). Such material originates from a mineral source. The authors performed an excellent characterization of the material in bulk form, using several electron microscopy and spectroscopic techniques that revealed the layered nature of this sulfide at atomic scale they were able to create an atomistic model of its crystalline structure. Furthermore the authors present the Raman spectra of an exfoliated crystal and show the characteristic peaks for bulk and monolayer regions.

The authors also present results for electrochemical characterization of basal plane and edge surface of the crystals, finding a maximum capacitance of $152.9 \mu\text{Fcm}^{-2}$. Similarly they performed experiments of electron transfer and hydrogen evolution reaction.

Lastly, the authors fabricated two field effect transistor devices that consisted of franckeite crystals as thin as 2 and 5 unit cells. The temperature dependence of the electron transport through these crystals revealed a semiconducting nature; p-type behavior was confirmed from the gating experiments. To the best of my knowledge data and methodology are valid and the materials characterization is presented in a staggering format.

While the quest for new atomically-thin layered materials is indeed important and the material characterization is remarkable good, in this report Vilecky et al. do not present a major scientific breakthrough. This situation is mainly because the authors did not probe any of the physical or chemical properties of a single layer (unit cell) of their material, it is well known that for the other 2D materials like graphene, TMDCs and phosphorene, the unprecedented physical properties are appreciated in the single layer form. Contrarily, the authors performed all the electrochemical measurement on bulk crystals and the transport experiments on few layer devices.

On the Raman spectra presented in figure 4, the monolayer region is smaller than the spot size (ca. 800 nm) hence there might be overlap between the monolayer and bulk areas.

The authors limited the comparison of the capacitance behavior of graphene and MoS₂ to their own experimental data, however there are reports where the capacitance of MoS₂ exceeds by orders of magnitude the one here reported ($152.9 \mu\text{Fcm}^{-2}$). See : Winchester, S Ghosh, S Feng, AL Elias, T Mallouk, M Terrones and S Talapatra, ACS applied materials & interfaces 6 (3), 2125-2130

In general I found this manuscript well written with a clear and engaging abstract, but the conclusions are not fully supported by the data, for instance the last sentence in the summary paragraph the authors state: "...these findings open the way to a new class of complex van der Waals crystals with prevalent basal cleavage as an alternative to artificially stacked 2D heterostructures 26" but along the manuscript they did not provide substantial evidence that this material would outperform artificially stacked 2D heterostructures as those reported in reference 26. Neither provide a path to control the properties of the material.

After considering the points presented above my recommendation is to Reject, however I also consider that a resubmission can be successful if this work is improved by accomplishing some of the following points:

- Large areas of the unit cell are consistently isolated and characterized
- The Raman spectra of single and few layer crystals are fully and properly indexed
- Photoluminescence spectra of the material depending on the number of layers is obtained
- Field effect devices are fabricated with single unit cell crystals
- A method for exfoliation (physical or chemical) of large quantities is presented
- A synthesis method to obtain extended area monolayers of the material is developed
- Band structure calculations for single layer, few layer and bulk are presented

Reviewer #2 (Remarks to the Author)

The authors report the exfoliation of franckeite crystals to single unit cell thickness facilitated by phase segregation. They observed the phase segregation of Pb/Sn anions and the formation of layered structures. The authors also physically exfoliated franckeite crystals into thin layers or even single unit cell thickness although the lateral size of the "single-unit-cell" area is very limited. This manuscript can be accepted by Nature Communications after several major revisions.

1. In the second paragraph of the main text, the authors claimed that the approximate chemical formula is $\text{Pb}_6\text{Sn}_3\text{Sb}_2\text{FeS}_{12}$, which is not charge balanced and does not match the generic chemical formula. The authors may need to measure the chemical composition more carefully.
2. In Figure 2c, the distribution of S along the vertical direction is not uniform, showing a periodic distribution. However, in Figure 2f, S shows an almost even distribution in Pb- or Sn-rich regions. The authors may need to confirm the element distribution of S measured by HAADF-STEM-EDXS method.
3. According to the XPS results, the surface chemical formula is $\text{Pb}_3\text{SnSb}_2\text{S}_6$, which is also not charge balanced. The author may need to point out the reason for this phenomenon. Surface defects? oxygen?
4. The authors may need to provide more detailed explanation for the Raman spectra of the bulky and thin-layered samples. The authors may also need to reach more conclusions based on the Raman measurements.
5. The authors claimed that they can get "single-unit-cell" thickness samples. However, the single-unit-cell area is too small to be applicable. The authors may need to try their best to exfoliate franckeite crystals into large-area thin layers, including several-layered and single-layered structures.

Reviewer #3 (Remarks to the Author)

The authors demonstrate the production of single unit cell thicknesses of franckeite by mechanical cleavage techniques, the success here being ascribed to a phase segregation into discrete layers. The thickness of the material is characterised by AFM, with Raman spectrometry, (S)TEM, -EDXS, cross-sectional imaging and XPS providing supporting materials characterisation. The authors demonstrate the potential for the use of franckeite in energy storage and conversion applications through electrochemical measurements of the electric double layer capacitance. In addition, FET characterisation of both a 2 layer and 5 layer franckeite device are presented, showing p doping.

The presented capacitances show an order of magnitude increase over those observed in graphene and MoS₂ for the basal plane, which the authors ascribe to the structure of electronic structure of franckeite and a likely additional contribution from pseudocapacitance that is described in the Supplementary Information.

Specific points:

The authors state that the structure of franckeite 'has not yet been fully determined' (page 5 line 2). The cited references span the years 1987-1995, and do not include more recent works, including those to be found at the RRUFF database <http://rruff.info/Franckeite> from 2002 and 2011. I suggest qualifying the above comment and citing these references where appropriate. The authors may or may not wish to adjust the title of the manuscript based on changes made here.

The authors state that of the cross-sectional sample presented in Fig 2e: 'the lower surface terminates partway through the Pb rich layer" (page 5 paragraph 2). Such a cleavage plane seems unlikely given the described structure - can the loss of Pb contrast be explained any other way?

The authors do not comment on the stability of the monolayer regions in ambient conditions except in the Methods section. Given the predicted band gap of the monolayer, the material is likely to be a candidate replacement for comparatively unstable black phosphorous monolayers, so some comment might serve to highlight this point.

Can the authors comment on the derived activation energies for the FET measurements of bilayer and 5 layer franckite with reference to the expected band gap?

Why do the authors not present FET characterisation of monolayer franckite presented in Fig 4a? Given that the isolation of monolayers is a core point of this manuscript, this should at least be addressed.

Overall, I believe that the isolation of a monolayer of the first naturally occurring van der Waals heterostructure material merits publication in Nature Communications, especially when supported by the extensive characterisation presented, although the above points should be addressed.

Reviewer #1:

In the manuscript titled: “Phase segregation facilitates exfoliation of franckeite crystals to a single unit cell thickness” by Matěj Velický and collaborators, a new atomically thin material is presented. The authors managed to isolate, by mechanical exfoliation, a single unit cell of a complex sulfide containing Lead (Pb), Tin (Sn), Antimony (Sb) and Iron (Fe), being Pb the one with highest atomic concentration (20 at %). Such material originates from a mineral source. The authors performed an excellent characterization of the material in bulk form, using several electron microscopy and spectroscopic techniques that revealed the layered nature of this sulfide at atomic scale they were able to create an atomistic model of its crystalline structure. Furthermore the authors present the Raman spectra of an exfoliated crystal and show the characteristic peaks for bulk and monolayer regions. The authors also present results for electrochemical characterization of basal plane and edge surface of the crystals, finding a maximum capacitance of 152.9 μFcm^{-2} . Similarly they performed experiments of electron transfer and hydrogen evolution reaction. Lastly, the authors fabricated two field effect transistor devices that consisted of franckeite crystals as thin as 2 and 5 unit cells. The temperature dependence of the electron transport through these crystals revealed a semiconducting nature; p-type behavior was confirmed from the gating experiments. To the best of my knowledge data and methodology are valid and the materials characterization is presented in a staggering format.

We are grateful for the reviewer’s comments and her/his praise for the quality of experimental characterisation and presentation of the results.

While the quest for new atomically-thin layered materials is indeed important and the material characterization is remarkable good, in this report Vilecky et al. do not present a major scientific breakthrough. This situation is mainly because the authors did not probe any of the physical or chemical properties of a single layer (unit cell) of their material, it is well known that for the other 2D materials like graphene, TMDCs and phosphorene, the unprecedented physical properties are appreciated in the single layer form. Contrarily, the authors performed all the electrochemical measurement on bulk crystals and the transport experiments on few layer devices. On the Raman spectra presented in figure 4, the monolayer region is smaller than the spot size (ca. 800 nm) hence there might be overlap between the monolayer and bulk areas.

Indeed, physicochemical properties of 2D materials monolayers are often distinctly different to those of the bulk form and we also attempted to show this for franckeite. We applied additional efforts to isolate monolayers of franckeite, which we indeed succeeded to do (revised Supplementary Fig. S7, S14). We obtained isolated franckeite monolayers (with no adjacent bulk crystals), so for these samples the Raman spectrum is indeed reliable (revised Fig. 4). We made attempts to measure transport properties and electrostatic force microscopy (EFM) on such monolayers – however, no successful results were obtained, which suggests that monolayers are probably unstable in ambient environment. Note that although monolayer franckeite appears stable when characterised by optical microscopy, AFM, and Raman, in

contrast, both transport and EFM are much more sensitive to the number of defects than the other techniques.

The thinnest confirmed stable structure is bilayer frackeite. We present Raman, transport and EFM measurements for such devices in the revised manuscript (Fig. 4, Fig. 6, and Supplementary Fig. S16) alongside with the Raman and AFM characterisation of monolayer frackeite. We would like to argue, however, that the electronic and structural properties do not depend on the number of layers. This is a result of the incommensurate stacking between T and H layers and is confirmed by the Raman measurements (the peaks are the same for monolayers, bilayers and thick samples). We added the related discussion to the revised manuscript.

The authors limited the comparison of the capacitance behavior of graphene and MoS₂ to their own experimental data, however there are reports where the capacitance of MoS₂ exceeds by orders of magnitude the one here reported (152.9 μFcm^{-2}). See : Winchester, S Ghosh, S Feng, AL Elias, T Mallouk, M Terrones and S Talapatra, ACS applied materials & interfaces 6 (3), 2125-2130.

This is an important issue in capacitance measurements, which is related to the discrepancy between the measured (apparent) and active (real) surface area. In our experiment, the measured surface area corresponds closely to the active surface area because of the small roughness of the mechanically exfoliated surface (root mean squared roughness was typically 0.5 – 1.0 nm, as stated in the original SI). In contrast, for the measurements using liquid-phase exfoliated MoS₂, as it is the case in the reference quoted by the reviewer (Winchester et al, ACS Appl Mat & Int 6, 2125, 2014), the exfoliated flakes are typically filtered and then compacted into a thin membrane-like electrode. This leads to significant increase in the real active surface area (and therefore specific capacitance in F cm^{-2}), due to the huge roughness and porosity of the re-stacked material. This overestimation often amounts to more than a few orders of magnitude difference. In fact, the authors of the discussed paper (Winchester et al), admit themselves that the exfoliation increases the area by at least 1 order of magnitude (second paragraph on the p.2127 in the paper: “*This also indicates that exfoliation results in an increase in the specific surface area of the MoS₂ (by at least 1 order of magnitude) because double-layer capacitance is proportional to specific surface area.*”).

Furthermore, in rough or porous materials, ions can diffuse into the interior of the bulk, and artificially inflate the measured capacitance. While such increase is indeed desirable for the end applications, i.e. supercapacitors, is it of little value for fundamental comparison of capacitance in different materials. We have now expanded the surface area/capacitance discussion in the revised manuscript accordingly.

In general I found this manuscript well written with a clear and engaging abstract, but the conclusions are not fully supported by the data, for instance the last sentence in the summary paragraph the authors state: ...these findings open the way to a new class of complex van der Waals crystals with prevalent basal cleavage as an alternative to artificially stacked 2D heterostructures 26” but along the manuscript they did not provide substantial evidence that this material would outperform artificially stacked 2D heterostructures as those reported in reference 26. Neither provide a path to control the properties of the material.

We see the importance of the paper in the demonstration of the naturally occurring complex van der Waals structures. Even though it is yet unclear if the particular structure will be used in applications, it serves as an important milestone in studying the principles of formation and stability of such complex crystals, which can be utilised when preparing the man-made heterostructures. In particular, the important findings of the current work are the role of polar

nature of the double-layer structure, charge transfer between the layers, and role of impurities, which balance the large charge transfer. We added this discussion to the main text.

After considering the points presented above my recommendation is to Reject, however I also consider that a resubmission can be successful if this work is improved by accomplishing some of the following points:

- *Large areas of the unit cell are consistently isolated and characterized*

In the revised manuscript we demonstrated successful, reproducible exfoliation of monolayer franckeite. We used such samples to perform Raman, transport and EFM measurements. As explained in the original manuscript and above, franckeite exfoliated to thin, needle-like flakes with high aspect ratio (typical length tens of micrometers, typical width less than 1 micrometer), this was consistent for various conditions of the mechanical exfoliation. We have repeated exfoliation of many more single- and few-layer crystals and confirmed that this large aspect ratio nature of the thin crystals is a reproducible phenomenon. Optical and AFM images of selected thin crystals have been added to the revised Supplementary Information (Supplementary Fig. S7, S14, 15).

- *The Raman spectra of single and few layer crystals are fully and properly indexed*

The tentative assignment of the individual metal sulphide Raman band has now been clarified in the revised manuscript. More accurate assignment and de-convolution of the individual scattering modes will require more involved theoretical and computational approach and it is currently beyond the scope of this work and our expertise.

- *Photoluminescence spectra of the material depending on the number of layers is obtained*

We have stated in the original manuscript that no significant photoluminescence was detected within the visible range. We have repeated these measurements multiple times using the newly exfoliated franckeite crystals and can now confirm that no photoluminescence is observed either on any flakes (monolayer, few-layer, or bulk) within the range of 532 – 900 nm, which is in agreement with the narrow infrared band gap determined from transport measurements and predicted from theory. We have now included the spectra in the Supplementary Information (Figure S9). We would also like to point out that since the half-layers are not commensurate, the individual layers are interacting only weakly, so one should not expect a strong dependence on the number of layers for any physical property.

- *Field effect devices are fabricated with single unit cell crystals*

We created a number of monolayer franckeite samples for Raman, transport and EFM measurements (revised Supplementary Fig. S7, S14, S16). Unfortunately, we found that monolayers are always insulating (most likely unstable), so we didn't manage to get transport data for a monolayer device. We did succeed in measuring transport data on bilayer devices, however. We extended our analysis of the transport data in the revised manuscript.

- *A method for exfoliation (physical or chemical) of large quantities is presented*

We have now performed sonication-assisted liquid-phase exfoliation of franckeite to demonstrate its feasibility as an alternative and scalable preparation method of this material. Liquid-phase exfoliation was tested using five different solvents (N-Methyl-2-pyrrolidone, dimethylformamide, acetone, isopropyl alcohol, and water). AFM characterisation shows that the exfoliated flakes tend to re-stack into large agglomerates even though thinner individual crystals with thickness less than 5 layers are still found. The above results and related discussion have now been included in the revised manuscript.

- *A synthesis method to obtain extended area monolayers of the material is developed*

As per the answer to one of the points above: The main importance of the paper is to demonstrate the existence and determine the properties of naturally occurring complex van der Waals structures. Even though it is yet unclear whether it will be possible to synthesise this particular structure and use it in applications, it serves as an important milestone in studying the principles of formation and stability of such complex heterostructures, which can be utilised when preparing the man-made heterostructures. This discussion has been included in the main text.

- *Band structure calculations for single layer, few layer and bulk are presented.*

We have performed the density functional theory (DFT) calculations of the half-layer, monolayer and few-layer franckeite and have now added the data to the revised manuscript and the Supplementary Information.

Reviewer #2:

The authors report the exfoliation of franckeite crystals to single unit cell thickness facilitated by phase segregation. They observed the phase segregation of Pb/Sn anions and the formation of layered structures. The authors also physically exfoliated franckeite crystals into thin layers or even single unit cell thickness although the lateral size of the "single-unit-cell" area is very limited. This manuscript can be accepted by Nature Communications after several major revisions.

We thank the reviewer for their positive feedback and acknowledge the comment on the small lateral size (width) of franckeite single unit cell flake, which are also shared with Reviewer #1.

1. In the second paragraph of the main text, the authors claimed that the approximate chemical formula is $Pb_6Sn_3Sb_2FeS_{12}$, which is not charge balanced and does not match the generic chemical formula. The authors may need to measure the chemical composition more carefully.

Admittedly, the chemical formula in the original manuscript was approximate and not charge-balanced. Two main factors affect the formula inaccuracy are:

- 1)** Low accuracy of EDXS quantification due to adsorption and geometric considerations especially in the absence of reference standards (Williams and Carter, Transmission Electron Microscopy, 2009). Quantification errors are greatest for the lightest elements and therefore their concentration is usually underestimated. This is likely to contribute to the apparent sulphur deficiency (excess positive charge) in comparison to the expected stoichiometry.
- 2)** Overlap between the S K α peak (2.31 keV used for quantification) and Pb M α peak at 2.35 keV. Although these two peaks were de-convoluted in the analysis, their overlap further reduces the accuracy of the quantification. We have now indexed the Pb M α peak in the EDXS spectrum of Fig. 1i.
- 3)** The relatively high concentration of impurity elements, including O (~4 at%), C (~8 at%), and Ag (~1 at%).

The mismatch between our formula and the generic literature formula of franckeite shown in the manuscript (which does not take the common impurities into consideration) is therefore largely caused by these three factors. We have now included oxygen in the formula (most of which is likely to be S substituent) and provided more accurate stoichiometric coefficients, resulting in the revised approximate chemical formula of $Pb_{6.0}Sn_{3.1}Sb_{2.5}Fe_{1.1}S_{12.0}O_{1.1}$. This formula is nearly charge-balanced, with ca. 5-10 % excess of positive charge (depending on the exact oxidation states of the metals).

We have amended the relevant section of the revised manuscript (second paragraph of the main text) and the Supplementary Information (section 1), and included a clear explanation based on the analysis above.

2. In Figure 2c, the distribution of S along the vertical direction is not uniform, showing a periodic distribution. However, in Figure 2f, S shows an almost even distribution in Pb- or Sn-rich regions. The authors may need to confirm the element distribution of S measured by HAADF-STEM-EDXS method.

We thank the reviewer for pointing this out. The quantified sulphur EDS data in Figure 2f does actually contain periodic distribution with a maximum of ~56% and a minimum of ~45%. However, the periodicity is not as obvious as in the case of the metallic elements due to similar reasons as for the point above:

- 1) EDXS detection and therefore quantification is known to be challenging for low atomic number elements due to the effects of absorption in the sample itself and in the detector.
- 2) The above-mentioned overlap between S K α and Pb M α peaks.
- 3) The fact that only small portion of the EDXS cross section (ca. 5 nm) is displayed. We have added some eye-guide lines to the graph to make correlation between metal-sulphur concentration peaks-troughs more obvious. We have also included a EDXS cross-sectional profile from a thick cross-section sample where the periodicity is clearly visible (revised Figure S3). We have now revised both the manuscript and SI according to the above discussion.

3. According to the XPS results, the surface chemical formula is Pb₃SnSb₂S₆, which is also not charge balanced. The author may need to point out the reason for this phenomenon. Surface defects? oxygen?

We are grateful to the reviewer for spotting this error, which we have now corrected in the revised manuscript. A mistake has been made when designing equation S4 used to calculate the stoichiometric coefficient v_i . The erroneous equation S4 described charge balance relationship between individual species, rather than the stoichiometry (this is the reason why the incorrect chemical formula Pb₃SnSb₂S₆ would be charge-balanced assuming all the individual elements had identical absolute charge).

The erroneous equation S4 has been replaced by the correct one - v_i is simply a fraction of the atomic percentage of element i ($v_i = x_i / \sum x_i$). The incorrect chemical formula has now also been replaced with a charge-balanced, accurate formula: Pb_{6.0}Sn_{2.0}Sb_{2.4}S_{13.8}O_{1.0}. The whole XPS analysis in the section 3 of the Supplementary Information thoroughly checked and corrected.

4. The authors may need to provide more detailed explanation for the Raman spectra of the bulky and thin-layered samples. The authors may also need to reach more conclusions based on the Raman measurements.

During the revision period we produced a number of additional monolayer, bilayer and few-layer samples, on which we systematically measure Raman, transport, and electrostatic force microscopy characteristics. Unlike in the previous manuscript version, the current Raman data were obtained on isolated flakes (with no adjacent bulk crystals) with uniform thickness, so we are 100% confident in our results. The new findings confirm our previous conclusions that the Raman spectrum does not strongly depend on the number of layers. We explain this being due to the fact that the T and H layers are not commensurate, therefore the multilayer stacking is not commensurate either, and hence the layers interact only weakly. The relevant sections of the manuscript have been revised accordingly.

5. The authors claimed that they can get "single-unit-cell" thickness samples. However, the single-unit-cell area is too small to be applicable. The authors may need to try their best to exfoliate francheite crystals into large-area thin layers, including several-layered and single-layered structures.

We have repeated exfoliation of francheite many time since, and we can now confirm that the exfoliation to thin, needle-like flakes with high aspect ratio (typical length tens of μm , typical width less than 1 μm) is a reproducible phenomenon, as we proposed in the original

manuscript, based on the TEM analysis. We have now highlighted these conclusions clearly in the revised manuscript and included additional optical and AFM images of these newly exfoliated thin crystals in the Supporting Information (Supplementary Figure S7 and S14).

Reviewer #3:

The authors demonstrate the production of single unit cell thicknesses of franckeite by mechanical cleavage techniques, the success here being ascribed to a phase segregation into discrete layers. The thickness of the material is characterised by AFM, with Raman spectrometry, (S)TEM, -EDXS, cross-sectional imaging and XPS providing supporting materials characterisation. The authors demonstrate the potential for the use of franckeite in energy storage and conversion applications through electrochemical measurements of the electric double layer capacitance. In addition, FET characterisation of both a 2 layer and 5 layer franckeite device are presented, showing p doping.

The presented capacitances show an order of magnitude increase over those observed in graphene and MoS₂ for the basal plane, which the authors ascribe to the structure of electronic structure of franckeite and a likely additional contribution from pseudocapacitance that is described in the Supplementary Information.

Specific points:

The authors state that the structure of franckeite 'has not yet been fully determined' (page 5 line 2). The cited references span the years 1987-1995, and do not include more recent works, including those to be found at the RRUFF database <http://rruff.info/Franckeite> from 2002 and 2011. I suggest qualifying the above comment and citing these references where appropriate. The authors may or may not wish to adjust the title of the manuscript based on changes made here.

Unfortunately, we indeed missed these important references and we thank the reviewer for pointing this out. We have now added the two most recent references as requested, and also added further relevant references within (some of these references were used in the construction and discussion of the newly added density functional theory calculations of the franckeite band structure – Supplementary Information, section S7).

The authors state that of the cross-sectional sample presented in Fig 2e: 'the lower surface terminates partway through the Pb rich layer' (page 5 paragraph 2). Such a cleavage plane seems unlikely given the described structure - can the loss of Pb contrast be explained any other way?

The termination partway through the Pb rich layer is indeed surprising. However, these crystals are known to be highly faulted and it is therefore likely that here cleavage may have been associated with a stacking fault which locally weakened the structure. This discussion is now included in the revised manuscript.

The authors do not comment on the stability of the monolayer regions in ambient conditions except in the Methods section. Given the predicted band gap of the monolayer, the material is likely to be a candidate replacement for comparatively unstable black phosphorous monolayers, so some comment might serve to highlight this point.

During the revision period we produced a number of additional monolayer, bilayer and few layers samples, on which we systematically measure Raman, transport, and electrostatic force microscopy (EFM) properties. We have already briefly commented on the stability of the exfoliated crystals in the experimental section of the originally submitted manuscript. The

stability of thin franckeite crystals in ambient environment (air, room temperature) was confirmed by optical microscopy, AFM, and Raman spectroscopy, as no observable degradation seems to occur. However, only bilayer and thicker samples did conduct in the EFM experiment. At the same time, bilayer samples were very stable and survived for a long time in the ambient environment. At this moment it is difficult to name the exact reason, why the monolayer samples did not conduct. One reason might be the breaking of the ideal stoichiometry at the surface, or another possible reason is the surface oxidation, as tentatively suggested by the TEM and XPS analysis.

We have now reorganised the discussion in the revised manuscript accordingly, in particularly in the context of other promising, but unstable 2D crystals, such as black phosphorus, mentioned by the reviewer.

Can the authors comment on the derived activation energies for the FET measurements of bilayer and 5 layer franckeite with reference to the expected band gap?

We have extracted the activation energies to be 80 meV for a 5 layer, 170 meV for a 4 layer, and 220 meV for a bilayer device. We expect that those energies represent the size of the Schottky barrier in our devices. In this sense, it is in reasonable agreement with the expected gap, which is around 250 – 350 meV from our DFT calculations (see the updated manuscript).

Why do the authors not present FET characterisation of monolayer franckeite presented in Fig 4a? Given that the isolation of monolayers is a core point of this manuscript, this should at least be addressed.

Unfortunately, although we succeeded in fabrication many new monolayer samples, we were not able to obtain any monolayer devices that would conduct. Only bilayer or thicker devices were repeatedly conductive. Currently we do not have a firm explanation why monolayer devices do not conduct (even though we could obtain Raman signal from monolayer devices), however, it might be related to the changes in stoichiometry or oxidation at the surface, as discussed above.

Overall, I believe that the isolation of a monolayer of the first naturally occurring van der Waals heterostructure material merits publication in Nature Communications, especially when supported by the extensive characterisation presented, although the above points should be addressed.

We hope that we have now satisfied the comments with the efforts towards additional experiments, theoretical calculations, and analyses, which we have undertaken to improve the quality of the manuscript, based on the reviewer's comments.

In addition to the above specific revisions, we have updated the title, abstract, and conclusions according to the changes made, and performed a careful check of the text and corrected the style and grammar throughout both the manuscript and Supplementary Information. We hope that by the careful consideration of reviewers' concerns and suggestions, the revised manuscript has significantly improved in quality and is now of acceptable standard for publication in *Nature Communications*.

Reviewers' Comments:

Reviewer #1 (Remarks to the Author)

After reviewing the authors response I find all the question raised in the revision of the original manuscript satisfactorily completed. I recommend this manuscript for publication in nature communications in its present form.

Reviewer #2 (Remarks to the Author)

The authors have addressed my questions and suggestions in the last review. Therefore, this manuscript can be accepted for publication in Nature Communications.

Reviewer #3 (Remarks to the Author)

I thank the authors for addressing so completely the points raised by myself and the other reviewers, and for the extensive extra efforts they have gone to in short order. The resulting changes to the manuscript are highly satisfactory, with the only key open question in my opinion being the stability and mechanism for the lack thereof in monolayers: surely an avenue for further research beyond the scope of the present manuscript. I have no hesitation in recommending the manuscript for publication.

Response to Referees:

Reviewer #1 (Remarks to the Author):

After reviewing the authors response I find all the question raised in the revision of the original manuscript satisfactorily completed. I recommend this manuscript for publication in nature communications in its present form.

Reviewer #2 (Remarks to the Author):

The authors have addressed my questions and suggestions in the last review. Therefore, this manuscript can be accepted for publication in Nature Communications.

Reviewer #3 (Remarks to the Author):

I thank the authors for addressing so completely the points raised by myself and the other reviewers, and for the extensive extra efforts they have gone to in short order. The resulting changes to the manuscript are highly satisfactory, with the only key open question in my opinion being the stability and mechanism for the lack thereof in monolayers: surely an avenue for further research beyond the scope of the present manuscript. I have no hesitation in recommending the manuscript for publication.

We are pleased that all 3 reviewers find our revisions satisfactory and do not require any further revisions.